# Dehydroisohispanolone as a Promising NLRP3 Inhibitor Agent: Bioevaluation and Molecular Docking

**DOI:** 10.3390/ph15070825

**Published:** 2022-07-02

**Authors:** Laura González-Cofrade, Irene Cuadrado, Ángel Amesty, Ana Estévez-Braun, Beatriz de las Heras, Sonsoles Hortelano

**Affiliations:** 1Departamento de Farmacología, Farmacognosia y Botánica, Facultad de Farmacia, Universidad Complutense de Madrid (UCM), Plaza Ramón y Cajal s/n, 28040 Madrid, Spain; lagonz11@ucm.es (L.G.-C.); icberrocal@ucm.es (I.C.); 2Departamento de Química Orgánica, Instituto Universitario de Bio-Orgánica Antonio González, Universidad de La Laguna, Avda, Astrofísico Francisco Sánchez 2, 38206 La Laguna, Tenerife, Spain; aarnesty@ull.es; 3Unidad de Terapias Farmacológicas, Área de Genética Humana, Instituto de Investigación de Enfermedades Raras (IIER), Instituto de Salud Carlos III, Carretera de Majadahonda-Pozuelo Km 2, 28220 Madrid, Spain

**Keywords:** dehydroisohispanolone, diterpene, NLRP3 inflammasome, interleukin-1β, caspase-1, pyroptosis

## Abstract

Dehydroisohispanolone (DIH), is a labdane diterpene that has exhibited anti-inflammatory activity via inhibition of NF-κB activation, although its potential effects on inflammasome activation remain unexplored. This study aims to elucidate whether DIH modulates NLR family pyrin domain-containing protein 3 (NLRP3) inflammasome in macrophages. Our findings show that DIH inhibited NLRP3 activation triggered by Nigericin (Nig), adenosine triphosphate (ATP) and monosodium urate (MSU) crystals, indicating broad inhibitory effects. DIH significantly attenuated caspase-1 activation and secretion of the interleukin-1β (IL-1β) in J774A.1 cells. Interestingly, the protein expressions of NLRP3, apoptosis-associated speck-like protein containing a CARD (ASC), pro-caspase-1 and pro-IL-1β were not affected by DIH treatment. Furthermore, we found that DIH pretreatment also inhibited the lipopolysaccharide (LPS)-induced NLRP3 inflammasome priming stage. In addition, DIH alleviated pyroptosis mediated by NLRP3 inflammasome activation. Similar results on IL-1β release were observed in Nig-activated bone marrow-derived macrophages (BMDMs). Covalent molecular docking analysis revealed that DIH fits well into the ATP-binding site of NLRP3 protein, forming a covalent bond with Cys415. In conclusion, our experiments show that DIH is an effective NLRP3 inflammasome inhibitor and provide new evidence for its application in the therapy of inflammation-related diseases.

## 1. Introduction

The innate immune system is one of the first lines of defense against infection and initiates the process of tissue repair. Its activation after recognition of pathogen-associated molecular patterns (PAMPs) or danger-associated molecular patterns (DAMPs) mediates the inflammatory response leading to the release of pro-inflammatory mediators by macrophages. In response to these exogenous microbial invasions and endogenous damage, a collection of mammalian cytoplasmic multiprotein complexes named inflammasomes are formed, playing a key role in the regulation of immunity and inflammatory diseases. NLRP3 is one of the most extensively studied inflammasome sensors as its aberrant activation plays crucial roles in the pathogenesis of several inflammatory diseases, including type 2 diabetes, psoriasis, colitis and some infectious diseases, among others [1,2,3,4]. Thus, the NLRP3 inflammasome has raised enormous interest as a potential target for the development of new therapeutic approaches.

Importantly, NLRP3 inflammasome complex activation requires two steps: a first signal or “priming”, mediated through NF-κB activation, inducing the synthesis and accumulation of inflammasome protein NLRP3, pro-IL-1β and pro-IL-18 expression; and a second signal specific to the inflammasome. This can be triggered by many diverse stimuli including bacteria pore-forming toxins such as Nigericin (Nig), Reactive Oxygen Species (ROS) and non-microbial signals such as monosodium urate (MSU) crystals or asbestos and silica, making NLRP3 the most versatile inflammasome [5,6,7]. Subsequently, caspase-1 activation mediated by the inflammasome leads to the maturation and secretion of pro-inflammatory cytokines IL-1β and IL-18 [8,9]. Furthermore, caspase-1 activation also induces a type of inflammatory programmed cell death called pyroptosis, after the cleavage of pore-forming gasdermin D (GSDMD) protein [10]. Pyroptosis is characterized by the loss of cell membrane integrity and the release of cytosolic lactate dehydrogenase (LDH).

Natural products, in particular diterpenes, have been identified as an important potential source of new pharmacological agents, having a broad spectrum of biological activities [11,12,13,14]. Hispanolone is a labdane diterpene first isolated from the aerial parts of *Ballota hispanica* (Labiatae). In previous studies, we have reported that hispanolone derivatives exhibited great therapeutic potential as anti-inflammatory, cardioprotective and anti-tumoral agents [15,16,17,18,19]. Of them, dehydroisohispanolone diterpene (DIH) showed potent anti-inflammatory activity, as indicated by transcriptional inhibition of pro-inflammatory enzyme Nitric Oxide Synthase-2 (NOS-2) and Cyclooxygenase-2 (COX-2) expression mediated by inhibition of nuclear translocation of NF-κB transcription factor in macrophages [20]. Nevertheless, no available data addressed the potential effects of the diterpene DIH on inflammasome modulation. Therefore, this study aimed to explore the impact of the diterpene DIH on NLRP3 inflammasome activation in macrophages.

## 2. Results

### 2.1. Cytotoxic Effects of DIH on Macrophages

The chemical structure of DIH (C_20_H_28_O_2_) is shown in Figure 1A. DIH was obtained from the natural diterpene hispanolone following the procedure described by Pérez-Sirvent et al. [21] and the corresponding ^1^H-NMR and ^13^C-NMR data are shown (Appendix A and Appendix A). To evaluate the effect of DIH on cell viability, J774A.1 cells were incubated with the indicated concentrations for 24 h and cytotoxic effects were determined using the MTT assay. As shown in Figure 1B, no significant changes in cell viability were observed in J774A.1 macrophages treated with DIH at concentrations up to 20 μM. Based on these results, DIH at a concentration of 10 and 20 μM was mostly used for further experimentation.

### 2.2. DIH Reduces IL-1β Secretion following NLRP3 Inflammasome Activation

Numerous reports have confirmed that activation of NLRP3 inflammasome triggers caspase-1 activation leading to the release of pro-inflammatory IL-1β. In order to investigate the effects of DIH on NLRP3 inflammation, IL-1β release was evaluated using ELISA and western blot analysis. J774A.1 cells were primed with LPS and stimulated with various NLRP3 activators, including Nig, ATP and MSU crystals. The inhibitory potential of DIH was evaluated by pretreatment of the cells with different concentrations of this compound for 30 min prior to incubation with inflammasome stimuli. The results showed that DIH was able to inhibit IL-1β release triggered by all the NLRP3 stimuli (Figure 2A–C). Dose-dependent effects were observed in cells treated with Nig, showing a half-maximal inhibitory concentration (IC_50_) of approximately 13.49 µM (Figure 2A). Similar results were obtained in cells stimulated with ATP or MSU (Figure 2B,C). Western blot analysis confirmed the reduction of cleaved IL-1β levels in culture supernatants by DIH treatment without effects on the expression of pro-IL-1β in cell lysates (Figure 2D–F). These data suggest that inhibition of NLRP3 inflammasome activation might be involved in the reduction of IL-1β release by DIH.

### 2.3. Caspase-1 Activation Is Inhibited by DIH Treatment

To further explore the inhibitory effects of DIH, we evaluated the cleavage of caspase-1 as an essential step in the release of IL-1β after NLRP3 inflammasome activation. We found that DIH significantly reduced caspase-1 activity in LPS-primed J774A.1 macrophages stimulated with the NLRP3 activators (Nig, ATP and MSU) (Figure 3A). Moreover, western blot analysis revealed that DIH treatment also blocked caspase-1 cleavage (Figure 3B), without affecting pro-caspase-1 expression.

### 2.4. DIH Attenuates Inflammasome-Dependent Pyroptosis

To determine whether DIH could reduce inflammasome-induced pyroptosis, we analyzed LDH release and the levels of GSDMD-N, the activated form of GSDMD. Figure 4A shows that the LDH release induced by LPS and Nig was significantly reduced by DIH treatment at 20 µM. Additionally, western blot data confirmed that LPS + Nig stimulation significantly increased the expressions of GSDMD-N in J774A.1 macrophages (Figure 4B), whereas DIH treatment reduced the expression of GSDMD-N, indicative of pyroptosis inhibition. Interestingly, these results suggest that DIH not only inhibits caspase-1 activation but also blocks macrophage pyroptotic cell death.

### 2.5. DIH Also Affects the Priming Step of Inflammasome Activation

As previously described, activation of NLRP3 inflammasome is a two-step process consisting of priming and triggering. NLRP3 inflammasome components expression was examined by qPCR and western blot to investigate deep into the mechanisms involved in the inhibitory effects of DIH. When macrophages were treated with DIH after LPS priming, there was no inhibition of pro-IL-1β expression or other inflammasome components (NLRP3, ASC, IL-18, or pro-caspase-1), demonstrating that the DIH-induced NLRP3 inflammasome inhibition was not due to decreased expression of NLRP3 or pro-IL-1β under these conditions (Figure 5A). Interestingly, the protein levels of NLRP3 and ASC (Figure 5B) and pro-IL-1β (Figure 2D) or procaspase-1 (Figure 3B) were also unaffected by DIH added after LPS stimulation.

In order to clarify whether DIH affected the priming step of inflammasome activation, macrophages were treated with DIH prior to LPS priming. As shown in Figure 6A, after subsequent Nig activation, a concentration-dependent inhibition of IL-1β maturation was observed, indicating that DIH regulates pro-IL-1β expression via NF-κB. Indeed, the protein expression of pro-IL-1β induced by LPS was reduced by DIH treatment (Figure 6B). Additionally, expression levels of well-known LPS-induced inflammatory mediators, including NOS-2, COX-2, TNF-α or IL-6 were downregulated by DIH (Figure 6C). Under these conditions, DIH also inhibited IL-18 expression, although the concentration required for this effect was higher than for inhibition of pro-IL-1β expression. Interestingly, DIH had minimal effects on NLRP3 and procaspase-1 and ASC expression (Figure 6B,C).

### 2.6. Secretion of Pro-Inflammatory IL-1β Is Also Inhibited by the Treatment of DIH in BMDMs

To gain insight into the effects of DIH on NLRP3 inflammasome activation, we evaluated its effects in primary mouse BMDMs. First, we analyzed the cytotoxicity of DIH using cell viability and proved that the doses used in previous experiments were not cytotoxic to these cells (Figure 7A). Moreover, when LPS primed-BMDMs were treated with DIH prior to challenging with Nig, a significant reduction in the quantity of IL-1β secreted was observed (Figure 7B), confirming that DIH exerts an inhibitory role on NLRP3 inflammasome activation. Indeed, these results are similar to those obtained with the NLRP3 inflammasome specific inhibitor, MCC950, in J774A.1 cells and BMDM as shown in Appendix A). Additionally, DIH also exhibited inhibitory effects on the priming step as shown in Figure 7C, where IL-1β secretion was inhibited by DIH added prior to treatment with LPS and Nig.

### 2.7. Docking Studies on NLRP3

Molecular docking has proven to be an efficient strategy to speed up the process of drug discovery, in addition to being a powerful tool for understanding different protein functions. Indeed, it is an effective strategy in structure-based virtual screening. In the present study, molecular docking was performed in order to examine the possible binding mode, as well as to confirm the anti-inflammatory activity of DIH via inhibition of NLRP3 inflammasome activation. As is well known, among the NLRs, NLRP3 is the most studied of the nucleotide-binding domain (NBD) family and leucine-rich repeat (LRR)-containing proteins (NLRs). Furthermore, NLRP3 is composed of an N-terminal pyrin domain (PYD) that is the effector domain for supramolecular complex formation; a central triple-ATPase domain called NACHT that mediates protein oligomerization upon activation; and a C-terminal LRR domain that may act as the signal sensor [22]. A set of conserved Cysteines present in the NACHT domain are important for inflammasome activation [23]. ADP-bound NLRP3 is thought to be in an autoinhibited state that is relieved by the action of various kinases. This includes phosphorylation of the activation loop upstream of the nucleotide-binding domain (NBD) and interaction with the mitotic serine/threonine kinase NEK7 [24,25,26,27]. NLRP3 can be activated by a wide range of stimuli, including ATP, K^+^ ionophores [28], heme [29], particulate matter [30], pathogen-associated RNA [31] and bacterial and fungal toxins and components [32,33]. In healthy tissues, the plasma membrane of living cells is impermeable to ATP and ATP remains intracellularly linked to metabolism. Thus, extracellular levels of ATP are low. However, in distressed or damaged tissues extracellular ATP concentrations rise and act as a potent danger signal through the activation of purine receptors in effector cells. The release of ATP into the extracellular milieu could be due to a lytic-passive release coupled with cell death or non-lytic regulated release. ATPase activity within the NACHT domain is also mandatory for NLRP3 inflammasome activation. NLRP3 actually needs to cleave ATP into ADP in its ATPase pocket in order to achieve the active conformation [34]. Whatever stimulus triggers NLRP3 inflammasome activation, conformational changes are required in the switch from the inactive resting conformation to the active structure that, in its turn, can trigger a downstream cascade. All of this suggests that blocking ATPase activity might be a feasible strategy with which to target the design of compounds that can inhibit those syndromes associated with inflammatory disorders.

In order to offer a better understanding of the significant inhibitory activity of DIH as well as to explore its possible binding mode, conventional (non-covalent) and covalent docking studies were carried out using the Glide software [35] on the reported crystal structure of NLRP3 NACHT domain in complex with the potent inhibitor MCC950 analog (PDB 7ALV). Thus, DIH was docked in the same binding site as the MCC950 analog and also in the nucleotide-binding site where ADP is bound in the closed conformation of the crystal structure.

An in-depth analysis of the non-covalent docking result showed that the best docking poses for DIH (docking score: −5.24 kcal mol^−1^) occupy less surface area than the cognate ligand at the NLRP3 ligand binding site. Furthermore, it can also be observed that there is only a single key hydrogen bond interaction for receptor affinity between the carbonyl group of DIH and the residue Ala 228 while observing that the number of bad contacts is much higher compared to those interactions observed by the cognate ligand. The highest docking scores were observed in the range from −3.51 to −5.24 kcal mol^−^^1^ whereas the inhibitor MCC950, used as a reference, showed a docking score value of −8.50 kcal mol^−^^1^, suggesting that these interactions are not so effective in providing a stabilizing effect to the protein-ligand complex formed.

In accordance with the above and taking into consideration that DIH has an α, β-unsaturated carbonyl moiety in its structure that could act as a Michael acceptor, a covalent docking was carried out at the ATP binding site. Michael acceptors are able to react with biological nucleophilic residues, such as cysteine, serine, lysine and histidine, found in proteins and thus form covalent bonds. These observations allow us to hypothesize that the NLRP3 inflammasome could be a sensitive target to be inhibited by DIH.

Thus, it can be seen how the DIH compound fits very well and it is located exactly in the same region where the nucleotide is located, occupying a large portion at the ATP binding site (Figure 8).

As we expected, the docking result showed that our compound binds covalently to NLRP3, forming a covalent bond between Cys415 and the β carbon of the α, β unsaturated carbonyl moiety. In addition, another important aspect that can be derived from the predicted pose of DIH is the ability to also form two new hydrogen bond interactions, one of them between the carbonyl group with an Arg167 residue and another hydrogen bond interaction formed by the oxygen of the furan ring with a Thr169 residue. A π-π-stacking interaction between the furan ring with a residue of Phe373 was also observed, as well as multiple potential hydrophobic interactions involving residues such as Ile 151, Ile 234, Tyr 168, Thr169, Arg 167, Tyr381, Phe373, Pro412, Leu413, Trp416 and HIE522 (Figure 9). Lastly, it is worth highlighting that the cdock affinity values calculated for the DIH compound (cdock affinity range of −8.04 to −8.83 kcal mol^−1^), when compared with the scores obtained for the bioactive diterpenoid oridonin [36] (cdock affinity −7.14 kcal mol^−1^), which was used as the reference compound in the in silico study, it is evident that our compound binds efficiently to the ATP binding site of the inflammasome. In conclusion, and in contrast to other Michael acceptors, our analysis has revealed that DIH forms a covalent bond with a cysteine residue in the NACHT domain, thereby potentially blocking the interaction between NLRP3 and NEK7, preventing a conformational change and avoiding the assembly and activation of the NLRP3 inflammasome through the inhibition of its ATPase activity.

## 3. Discussion

Bioactive natural products are very promising molecules for the development of new therapeutic agents. Terpenes constitute one of the groups with great potential due to their wide range of biological activities [12,14,15,37]. In this context, labdane diterpenes, including hispanolone derivatives such as dehydroisohispanolone (DIH), have been reported to exert anti-inflammatory effects [15]. These compounds inhibit NF-κB binding to DNA, thus reducing the expression of pro-inflammatory proteins, such as COX-2 and NOS-2 [20,38,39,40]. In this study, we expand our research on the anti-inflammatory properties of DIH, focusing on its potential effects as a modulator of the inflammasome pathway. Inflammasomes are important regulators of immunity and inflammatory diseases, making NLRP3 one of the most attractive targets for the development of innovative therapeutic approaches due to its association with several human inflammatory and autoimmune diseases [1,2,3,4]. So far, the only therapeutic approach clinically available against inflammasome activation is the IL-1β signaling blockage using biologic agents [41]. However, IL-1β release is not the only resulting event following NLRP3 inflammasome activation. Indeed, caspase-1, as a critical component of the NLRP3 inflammasome, also induces IL-18 secretion and cleaves gasdermin D to induce pyroptosis [10], two processes that may contribute to disease progression. Nonetheless, most of the reported caspase-1 inhibitors are peptides with limited clinical applications [42,43]. Thus, the identification of new small molecules that directly target the NLRP3 inflammasome pathway, rather than IL-1β blockage, or the development of new non-peptidic inhibitors with effects on caspase-1 activity, are desirable and would result in increased efficacy and clinical application. 

In this study, the diterpene DIH was found to inhibit the NLRP3 inflammasome, as well as the cleavage of caspase-1 and the IL-1β release in LPS-primed macrophages triggered by multiple NLRP3 inflammasome activators (ATP, Nig and MSU), indicating broad inhibitory effects. Pyroptotic cell death triggered by caspase-1 activation was also inhibited by DIH. Similarly, other diterpenes and derivatives, also suppress inflammation through NLRP3 inhibition [36,44,45,46,47,48]. Additionally, these compounds also act by directing caspase-1 activity and suppressing downstream NLRP3-inflammasome activation-driven events.

Previous studies by our group have shown that DIH exhibited potent anti-inflammatory activity by targeting the NF-κB signaling pathway [20]. This current work confirmed these results and found that DIH also affected the LPS-priming signaling, as indicated by suppression of IL-1β expression when cells were treated with DIH before LPS stimulation and by a greater inhibition of IL-1β release under this treatment condition. The decrease in IL-1β production observed by DIH treatment after LPS challenge was not due to changes in gene expression or IL-1β precursor synthesis, suggesting that DIH effects might also be attributed to the inhibition of the NLRP3 inflammasome activation process. Taken together, the results obtained suggest that the anti-inflammatory effects of DIH are mediated by inhibition of the NLRP3 and NF-κB pathways synergistically (Figure 10). Therefore, the novelty of this study is that DIH also suppressed NLRP3 activation, exerting a dual inhibitory activity. Interestingly, our results are in line with those reported with other diterpenes such as oridonin and andrographolide that have shown anti-inflammatory effects via attenuation on both the NF-κB and NLRP3 signaling pathway [49,50].

To further find out the ability of DIH to directly target NLRP3 as a previous step for caspase-1 activation, we investigated the potential binding mode of DIH with NLRP3. A covalent molecular docking study of DIH onto the ATP-binding site (PDB 7ALV ) revealed that DIH binds covalently to NLRP3, forming a covalent bond with Cys415. Since the ATP-binding pocket of NLRP3 is essential for NLRP3 oligomerization and activation [34], our data might indicate that DIH inhibits activation of the NLRP3 inflammasome, possibly by inhibiting its ATPase activity.

## 4. Materials and Methods

### 4.1. Materials

Dehydroisohispanolone (DIH) (Figure 1A) was obtained from the acid treatment of natural hispanolone [21]. Thus, to 3.0 g (9.45 mol) of hispanolone in 175 mL of EtOH was added 10 mL of concentrated HCl, and the reaction mixture was heated under reflux for 18 h. Next, this was treated with 100 mL of H_2_O and extracted with CH_2_Cl_2_ (3 × 30 mL). The organic phases were collected, dried over anhydrous MgSO_4_, filtered and the solvent was removed. The residue was purified using column chromatography with hexanes/EtOAc (95:5) to yield 0.14 g (1.0%) of DIH and 1.3 g of dehydrohispanolone (46%) as yellow oils. DIH presented identical spectroscopic data to those reported [17]. The NLRP3 inflammasome inhibitor, MCC950 was obtained from InvivoGen (Toulouse, France).

### 4.2. Cell Cultures and Inflammasome Stimulation

J774A.1 murine macrophage cells were purchased from American Type Cell Culture (ATCC, Manassas, VA, USA) and were cultured in Dulbecco’s modified Eagle medium (DMEM, Sigma-Aldrich, Sant Louis, MI, USA) containing 10% fetal bovine serum (FBS), 100 U/mL penicillin and 100 μg/mL streptomycin, at 37 °C in a humidified incubator containing 5% CO_2_.

Primary mouse bone marrow-derived macrophages (BMDMs) were collected from the tibia and femur of C57BL/6 mice and cultured for 7–10 days in DMEM supplemented with 10% FBS, 100 U/mL penicillin, 100 μg/mL streptomycin and 30% of L929 mouse fibroblast-conditioned media.

For inflammasome activation, cells (1 × 10^6^) were cultured in 6-well plates, incubated overnight in a complete medium and later were changed to DMEM (1% FBS). Cells were primed for 4 h with 1 μg/mL LPS (from Escherichia coli O55:B5, Sigma) and DIH at different concentrations was added into the culture for 30 min. After that, cells were stimulated for 30 min with ATP (5 mM, Sigma), for 1 h with Nigericin (Nig) (20 mM, Sigma) or for 24 h with MSU (150 μg/mL, Sigma).

### 4.3. Cytotoxicity Assay

Cell viability was measured using a colorimetric 3-(4,5-dimethylthiazol-2-yl)-2,5-diphenyltetrazolium bromide (MTT) assay as previously described [42]. Briefly, J774A.1 cells (2 × 10^4^) and BMDMs (10^5^) were plated in 96-well plates and incubated in the presence of different concentrations of DIH for 24 h. MTT was then added and the plates were incubated at 37 °C for an additional 3 h. The reaction product, formazan, was extracted with dimethyl sulfoxide (DMSO) and the absorbance was read at 570 nm. Assays were performed in triplicate and results are expressed as the percentage of cell viability compared to untreated control cultures for at least three independent experiments. 

### 4.4. IL-1β Measurement

Supernatants from cell cultures were collected and IL-1β levels were determined using a mouse IL-1β ELISA kit (Mouse IL-1β/IL-1F2 DuoSet ELISA, R&D Systems, Minneapolis, MN, USA, DY101), according to the manufacturer’s instructions.

### 4.5. Caspase-1 Activity Assay

Caspase-1 activity was assayed using bioluminescent or fluorescence methods according to the stimuli. For LPS + Nig or LPS + ATP stimulated macrophages, Caspase-Glo 1 inflammasome Assay kit (Promega, Madison, WI, USA, G9951) was used according to the manufacturer’s instructions. For LPS + MSU stimulated macrophages, caspase-1 activity was measured with the fluorogenic substrate Z-YVAD-AFC (ALX-260-035-M001). Fluorescence of the AFC released from the Z-YVAD-AFC substrate was measured by an increase in fluorescence (λex = 400 nm, λem = 505 nm.). Relative luminescence/fluorescence units were transformed into percentage change, setting 100% for LPS + stimuli treatment. For each concentration of DIH, the percentage change was calculated.

### 4.6. Lactate Dehydrogenase (LDH) Assay

LDH release into the culture medium was determined by CytoTox 96 Non-Radioactive Cytotoxicity Assay (Promega, G1780), according to the manufacturer’s instructions.

### 4.7. RNA Extraction and Real-Time PCR

RNA was isolated from cells with Trizol reagent (Invitrogen, Waltham, MA, USA) and reverse transcription to cDNA was performed using M-MLV reverse transcriptase (Invitrogen, 28025-013). Quantitative PCR (SYBRgreen) analysis was performed with a StepOnePlus System (Applied Biosystems, Waltham, MA, USA). All samples were run in duplicate and 36B4 (acidic ribosomal phosphoprotein P0) was used as an endogenous control for normalization of the expression level of target genes. Relative quantification of gene expression was calculated using the 2-ΔΔCT method. Primer sequences are shown in Appendix A.

### 4.8. Immunoblot Analysis

J774A.1 cells were lysed using RIPA buffer, containing 0.5% Chaps, 10 mM Tris pH 7.5, 1 mM EGTA, 1 mM MgCl_2_, 10% glycerol and 5 mM β-mercaptoethanol, supplemented with phosphatase and protease inhibitor cocktails (Sigma). Protein content was assayed with the PierceTM BCA Protein Assay Kit (Thermo Scientific, Waltham, MA, USA) following the manufacturer’s instructions. Cell lysates and supernatant proteins were separated by SDS-PAGE gel electrophoresis, transferred onto PVDF membranes (Millipore, Burlington, MA, USA) and proved with the antibodies shown in Appendix A.

Blots were developed with Immobilon Western Chemiluminescent HRP Substrate according to the manufacturer’s instructions (Millipore) and visualized with the ChemiDoc Imaging Systems (Chemidoc XRS+, Bio-Rad, Hercules, CA, USA.).

### 4.9. Protein Preparation and Molecular Docking 

Docking studies were performed using Glide v9.1 The structure of NLRP3 (PDB 7ALV) was prepared for docking using the Protein Preparation Workflow (Schrodinger, LLC, New York, NY, USA, 2021) accessible from the Maestro program (Maestro, version 12.8; Schrodinger, LLC: New York, NY, USA, 2021). Bond corrections were applied to the cocrystallized ligand and an exhaustive sampling of the orientations of groups was performed.

Finally, the receptor was optimized in Maestro 12.8 using OPLS4 force field before the docking study. In the final stage, the optimization and minimization of the ligand-protein complexes were performed and the default value for RMSD of 0.30 Å for non-hydrogen atoms was used. The receptor grids were generated using the prepared protein, with the docking grids centered at the bound ligand. A receptor grid was generated using a 1.00 van der Waals (vdW) radius scaling factor and 0.25 partial charge cutoff. The ADP binding site was enclosed in a grid box of 20 Å^3^ without constraints. The three-dimensional structures of the ligand to be docked were generated and prepared using LigPrep as implemented in Maestro 12.8 (LigPrep, Schrodinger, LLC: New York, NY, USA, 2021) to generate the most probable ionization states at pH 7 ± 1 (retaining the original ionization state). At this stage, a series of treatments were applied to the structure. Finally, the geometries were optimized using OPLS4 force field. These conformations were used as the initial input structures for the docking. The ligand was docked using the extra precision mode (XP) [51] without using any constraints and a 0.80 van der Waals (vdW) radius scaling factor and 0.15 partial charge cutoff. The dockings were carried out with flexibility of the residues of the binding site near to the ligand. The generated ligand poses were evaluated with an empirical scoring function implemented in Glide, GlideScore, which was used to estimate binding affinity and rank ligands [52]. The XP Pose Rank was used to select the best-docked pose for each ligand.

### 4.10. Covalent Docking

Covalent docking studies were performed using the module CovDock workflow as implemented in Schrödinger Suite 2021-2 version. The CovDock workflow has been reported to be highly accurate in pose prediction of covalent inhibitors [53]. The prepared ligands were selected from the project table and were confined to the box, which was the centroid of ADP. The enclosing box size was calculated automatically based on the size of ADP. The residue Cys415 was defined as reactive residue from the crystal structure of NLRP3 NACHT (PDB 7ALV) domain in complex with an inhibitor on the workspace. The customized covalent docking algorithm was then selected as the reaction type. No constraints were imposed on the ligand for docking and pose prediction mode was selected. A total energy of 2.5 kcal mol was set as the cutoff to retain poses for further refinement by default, while the maximum number of poses to retain for further refinement was 200. The output poses per ligand reaction site was set to 10 best poses and the scoring option MM-GBSA was selected in order to obtain more information about the binding affinity of the poses and the best scoring six poses were analyzed.

### 4.11. Statistical Analysis

Statistical analyses were carried out using GraphPad Prism (version 9). Data are presented as means ± standard deviation (SD) from at least three experiments and a one-way ANOVA was performed. *p* < 0.05 was considered statistically significant. * *p* < 0.05, ** *p* < 0.01, *** *p* < 0.001. IC_50_ estimated value for DIH on IL-1β secretion was calculated using SigmaPlot software.

## 5. Conclusions

The results of the present study indicate that DIH suppresses the activation of caspase-1 and the subsequent secretion of IL-1β associated with inhibition of the NLRP3 inflammasome in macrophages. The compound also adopted a favorable conformation in the ATP binding pocket of NLRP3 showing the formation of a covalent link with Cys415, thus suggesting that DIH exerts its inhibitory activity by directly targeting NLRP3. We also report that DIH can target NLRP3 to exert its anti-inflammatory activity acting as a dual inhibitor of both NF-κB and NLRP3 and decreasing pyroptotic cell death. The combined inhibitory effects of DIH on both the priming step and the NLRP3 inflammasome activation process suggest that DIH may have therapeutic potential for the treatment of numerous inflammatory diseases. Moreover, our findings improve the understanding of the anti-inflammatory effect mediated by DIH via suppression of NLRP3 inflammasome.

## Figures and Tables

**Figure 1 pharmaceuticals-15-00825-f001:**
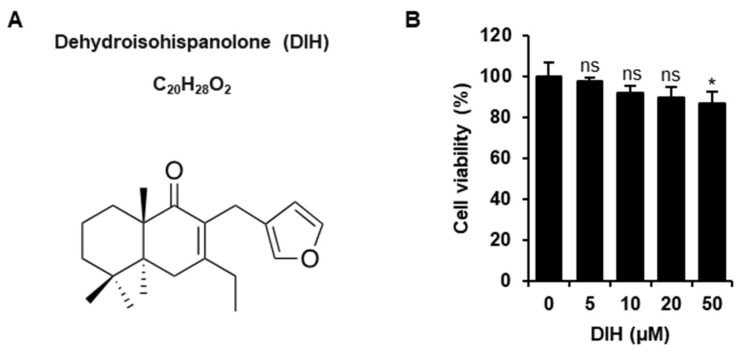
Chemical structure and cytotoxic effects of DIH on macrophages. (**A**) Chemical structure of dehydroisohispanolone (DIH) (**B**) J774A.1 cell viability after treatment with DIH (5–50 μM) for 24 h was determined by MTT assay and results are reported as mean ± SD (*n* = 3). ns = not significant and * *p* < 0.05 vs. untreated cells.

**Figure 2 pharmaceuticals-15-00825-f002:**
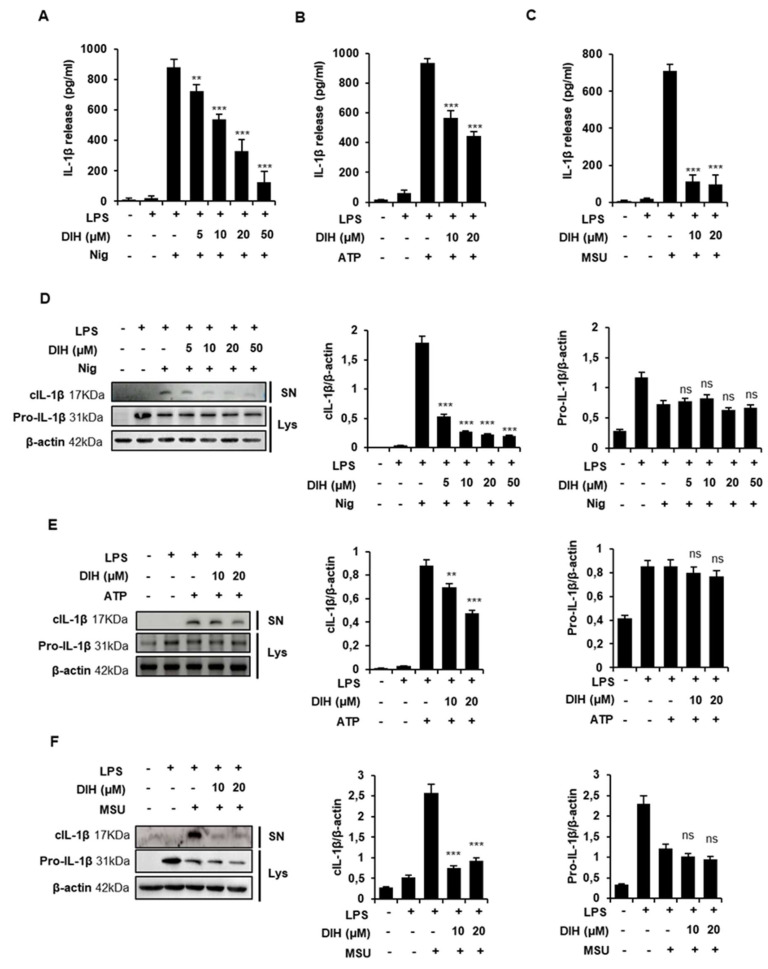
DIH reduces IL-1β secretion following NLRP3 inflammasome activation. (**A**,**D**) J774A.1 macrophages were primed for 4 h with LPS (1 μg/mL), followed by treatment with various doses of DIH (5, 10, 20, 50 μM) for 30 min and then Nig (20 μM) for 1 h. (**B**,**E**) LPS-primed macrophages were treated with 10 or 20 μM of DIH and then stimulated with ATP (5 mM, 30 min). (**C**,**F**) LPS-primed macrophages were treated with 10 or 20 μM of DIH and then stimulated with MSU (150 μg/mL, 24 h). (**A**–**C**) Levels of IL-1β in the culture medium were measured using ELISA. Results show the means ± SD of three independent experiments. (**D**–**F**) Supernatants (SN) and cell lysates (Lys) were analyzed by immunoblot of cleaved IL-1β (17 kDa) and pro-IL-1β (31 kDa) expression. β-actin was immunoblotted as a loading control. Densitometry analysis shows the mean ± SD of the western blot experiments. ns = not significant, ** *p* < 0.01 and *** *p* < 0.001 vs. LPS + stimuli (Nig, ATP or MSU).

**Figure 3 pharmaceuticals-15-00825-f003:**
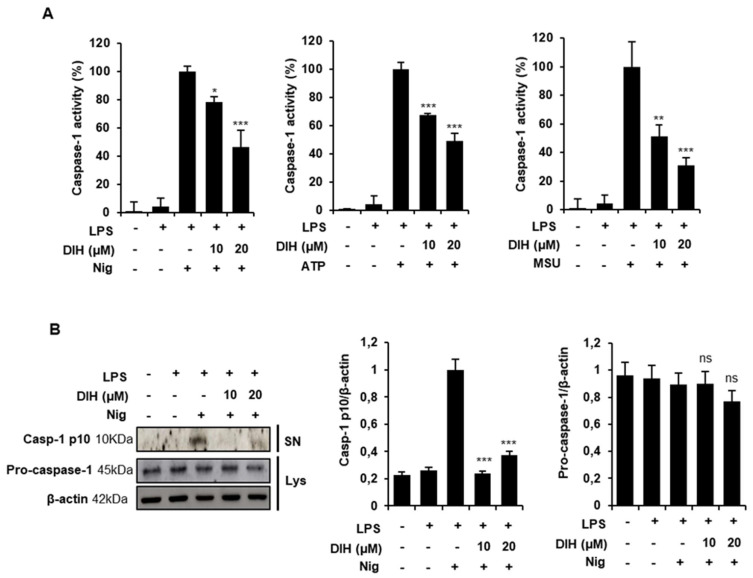
DIH suppresses Caspase-1 activation. (**A**) Caspase-1 activity was determined in LPS-primed J774A.1 cells stimulated with the inflammasome activators: Nig (20 μM, 1 h), ATP (5 mM, 30 min) or MSU (150 mg/mL, 24 h) in the presence of DIH (10, 20 μM). Data are expressed as means ± SD of percentage of caspase-1 activity in three independent experiments. (**B**) Cleaved caspase-1 (10 kDa) and pro-caspase-1 (45 kDa) expression were analysed by western blot in supernatants (SN) and cell lysates (Lys) after LPS + Nig stimulation in presence of DIH (10, 20 μM). β-actin was immunoblotted as a loading control. Densitometry analysis shows the mean ± SD of the western blot experiments. ns = not significant, * *p* < 0.05, ** *p* < 0.01 and *** *p* < 0.001 vs. LPS + NLRP3 activator treatment.

**Figure 4 pharmaceuticals-15-00825-f004:**
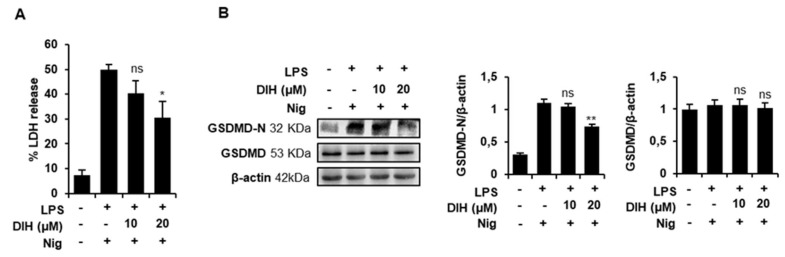
DIH attenuates NLRP3-dependent pyroptotic cell death. (**A**) LDH release was measured using a CytoTox^®^ kit in the supernatants of LPS-primed J774A.1 macrophages treated with DIH (10, 20 µM) and stimulated with Nig (20 µM, 1 h). Results are expressed as means ± SD (n = 3). (**B**) Immunoblot analysis of GSDMD-N and GSDMD expression in cell lysates. β-actin was immunoblotted as a loading control. Densitometry analysis shows the mean ± SD of the western blot experiments. ns = not significant, * *p* < 0.05 and ** *p* < 0.01 vs. LPS + Nig treatment.

**Figure 5 pharmaceuticals-15-00825-f005:**
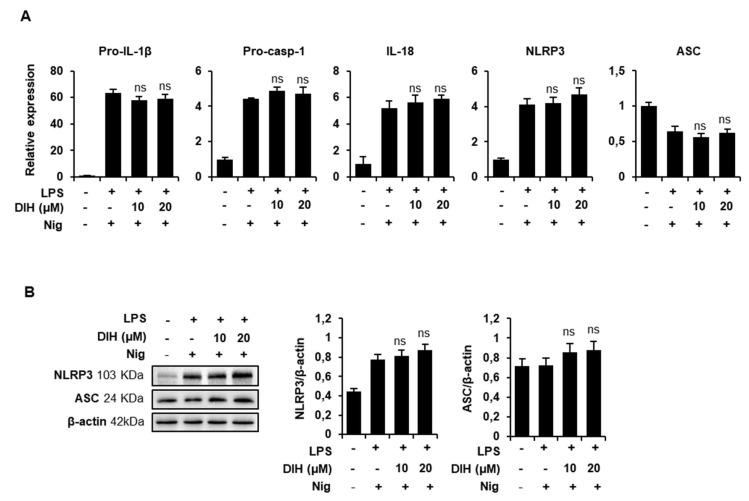
DIH has no effects on NLRP3 inflammasome components. J774A.1 macrophages were primed with LPS (1 μg/mL), then treated with DIH (10, 20 µM) and then stimulated with Nig (20 µM, 1 h). (**A**) Fold change of mRNA levels of inflammasome complex components. Results are expressed as means ± SD (n = 3). (**B**) Cell lysates after treatments were determined by immunoblot analysis of NLRP3 (103 kDa) and ASC (24 kDa) expression. β-actin was immunoblotted as a loading control. Densitometry analysis shows the mean ± SD of the western blot experiments. ns = not significant vs. LPS + Nig treatment.

**Figure 6 pharmaceuticals-15-00825-f006:**
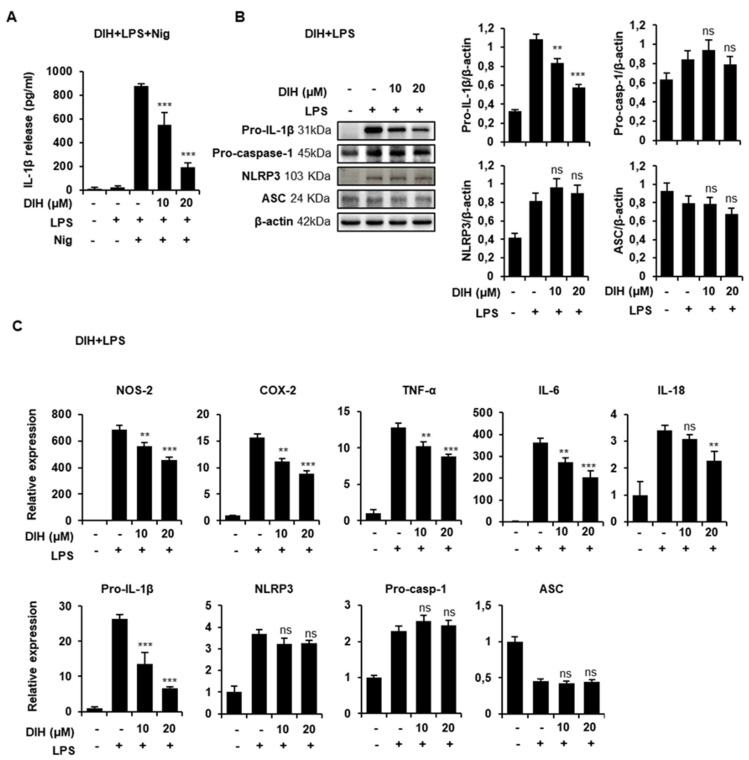
DIH regulates the priming step of NLRP3 inflammasome activation. (**A**) Levels of IL-1β were measured using ELISA in the culture medium from J774A.1 cells stimulated with LPS (1 μg/mL, 4 h), followed by treatment with DIH (10, 20 μM, 30 min) and then Nig (20 μM, 1 h). Results show the means ± SD (n = 3). *** *p* < 0.001 vs. LPS + Nig treatment. (**B**) J774A.1 were preincubated with DIH (10, 20 μM, 30 min) and then stimulated with LPS (1 μg/mL, 4 h). Relative expression of mRNA levels of inflammatory mediators and inflammasome complex components were measured by qPCR. Results are expressed as means ± SD of three independent experiments. (**C**) Immunoblot analysis of cell lysates for pro-IL-1β (31 kDa), pro-caspase-1 (45 kDa), NLRP3 (103 kDa) and ASC (24 kDa) expression, using β-actin as a loading control. Densitometry analysis shows the mean ± SD of the western blot experiments. ns = not significant, ** *p* < 0.01, *** *p* < 0.001 vs. LPS treatment.

**Figure 7 pharmaceuticals-15-00825-f007:**
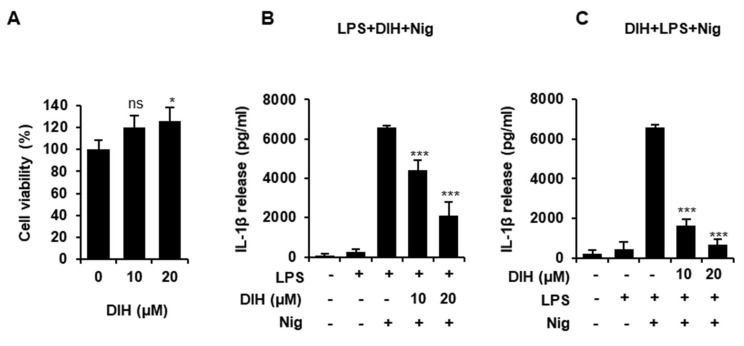
DIH reduces NLRP3 inflammasome activation in BMDMs (**A**) BMDMs viability after treatment with DIH (10, 20 μM) for 24 h was determined by MTT assay and results are reported as mean ± SD (n = 3). ns = not significant and * *p* < 0.05 vs. untreated cells. (**B**) BMDMs were primed for 4 h with LPS (1 μg/mL), followed by treatment with DIH (10, 20 μM) for 30 min and then Nig (20 μM) for 1 h. (**C**) BMDMs were preincubated with DIH (10, 20 μM) for 30 min, then stimulated with LPS (1 μg/mL, 3 h) and then with Nig (20 μM) for 1 h. Levels of IL-1β in the culture medium were measured using ELISA. Results show the means ± SD of three independent experiments. *** *p* < 0.001 vs. LPS + Nig treatment.

**Figure 8 pharmaceuticals-15-00825-f008:**
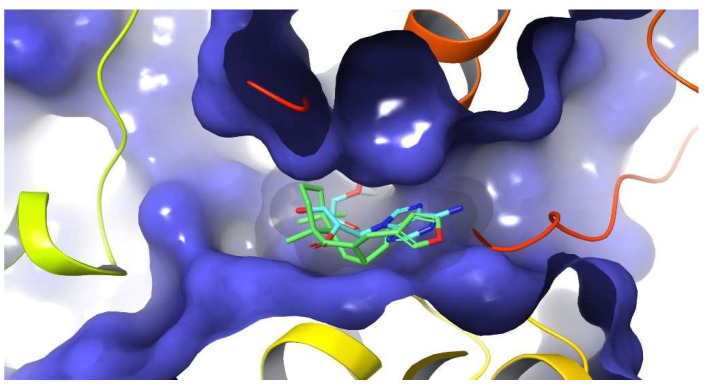
Covalent Docking of DIH occupying a large part of the ATP binding site (PDB 7ALV).

**Figure 9 pharmaceuticals-15-00825-f009:**
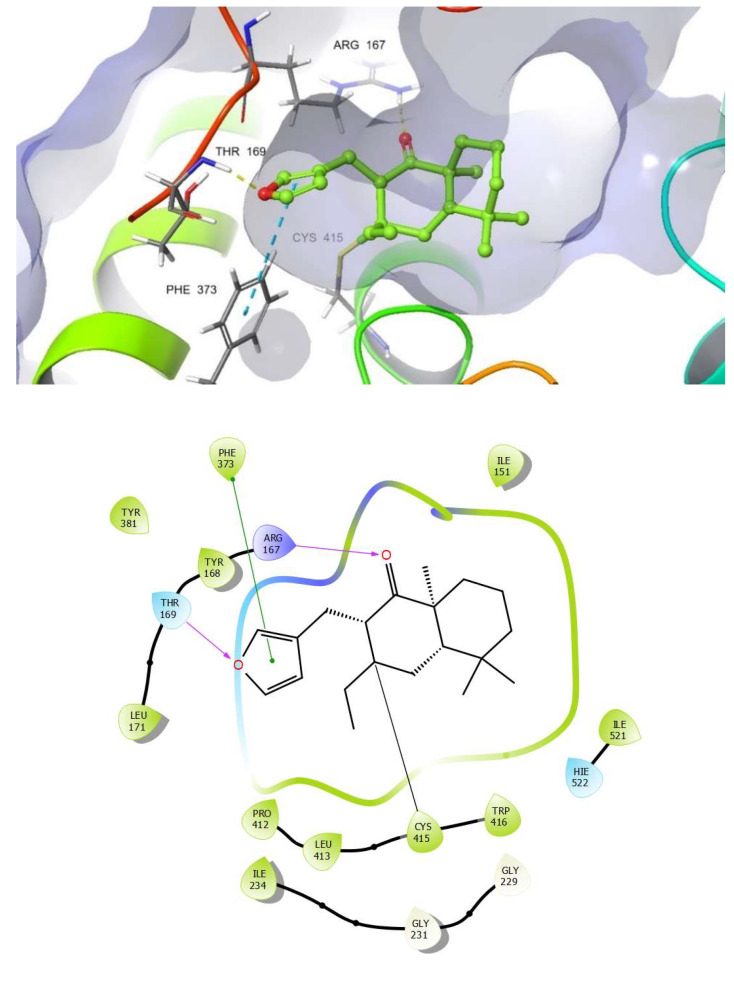
Covalent docking of DIH into the ADP binding site (PDB 7ALV) and its key interactions.

**Figure 10 pharmaceuticals-15-00825-f010:**
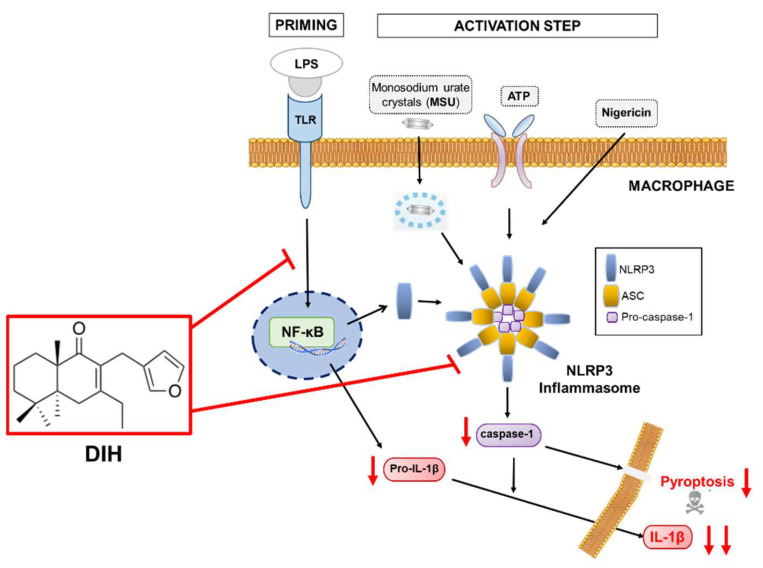
Molecular inhibitory mechanisms of DIH in the NLRP3 inflammasome pathway. Red down arrows indicate the different steps affected by DIH.

## Data Availability

Data is contained within the article and Appendix A.

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
