# Peer review of "Dehydroisohispanolone as a Promising NLRP3 Inhibitor Agent: Bioevaluation and Molecular Docking"

_pharmaceuticals, 2022, doi:10.3390/ph15070825_

Round 1

Reviewer 1 Report

The manuscript entitled "Dehydroisohispanolone as a promising NLRP3 inhibitor agent. Bioevaluation and Molecular Docking" by Laura Gonzalez-Cofrade et al. focuses on the potential application of DIH in the therapy of inflammation-related diseases as a NLRP3 modulator. The topic is important and interesting, and in my opinion it fully deserves publication in Pharmaceuticals. However, I have some minor comment which should be taken into account before acceptance of the present paper: While the presented docking studies are sound, I miss in this part of the manuscript a comparison to some reference drug. Please, add some discussion for a reference substance to complete this part of the manuscript.

Author Response

Following the reviewer´s suggestion, we have expanded the section regarding the docking study. In the revised version, a covalent docking study of DIH into the ATP biding site of NLRP3 has been included. Furthermore, we have compared the results of DIH with those obtained with the inhibitor MCC959 and also with the covalent inhibitor oridonin.

In the attachement file, point by point response to all reviewers is included.

Reviewer 2 Report

pharmaceuticals-1767852, Dehydrohispanolone-derived diterpene as a promising NLRP3 inhibitor agent. Bioevaluation and Molecular Docking

The manuscript presents a very good research. The methodology is correctly chosen and the results seem to be analyzed in a proper manner. I did not find any evident major problems to point out.

I have some problems with the docking analysis. I’m not familiar with the structure of NLRP3. Looking for details on the subject I found the following article.

https://www.ncbi.nlm.nih.gov/pmc/articles/PMC7479093/

Here the authors highlighted the importance of some cysteine residues. Considering the structure of the Dehydrohispanolone I would hypothesized a Michael addition of a thiol group from a cysteine residue to the carbon double bond of the compound.

I advise the authors to describe better the structure of NLRP3 in the docking section. Also, I would consider a covalent docking study towards the mentioned cysteine residues. Discuss if the experimental data point out to a covalent binding mechanism or not.

I’m not familiar with the structure of NLRP3, but I guess that it would be ATP on row 232, and not ATP. The authors should also dock ATP and comment the results.

In the discussion section, the authors should add more information (if available) on the toxicity of the compound. Point out that it has a known toxicophore, the α,β-unsaturated ketone.

The docking study in not integrated with the experiments. Are the results correlated?

The manuscript has some editing problems that could be easy corrected applying the mdpi styles in the whole the document.

Author Response

Following the reviewer´s suggestion, we have carried out a covalent docking study in order to test the possibility of DIH could act as a Michael acceptor of a thiol group of a cysteine residue.

These studies have been carried out using the reported NLRP3 crystal structure (PDB 7AVL) instead of (PDB 6NPY) since the data concerning to NLRP3 structure are more recent and because they include crystallographic data with the noncovalent NLRP3 inhibitor MCC950 analog.

Thus, DIH was docked in the same binding site that MCC950 analog, but low values of docking score were obtained. When covalent docking was carried out, it was found a covalent bond of DIH with Cys415. For these reason, we think that the DIH action mode could be by inhibiting its ATPase activity. 

 The docking study in not integrated with the experiments. Are the results correlated?

 Experimental data show that DIH significantly inhibits not only IL-1β release and pyroptosis but also caspase-1 activity following NLRP3 inflammasome activation. These effects are independent on the stimuli used (ATP, Nig or MSU), indicating broad inhibitory effects. Moreover, they suggest that DIH may be targeting NLRP3 as a common step for every stimuli and as the previous step for caspase-1 activation. Therefore, we investigated the ability of DIH to target directly NLRP3 performing docking studies to examine the potential binding mode of DIH. Molecular docking study suggest that fits into the ATP-binding pocket of NLRP3, which is essential for its oligomerization and activation. Thus, DIH may be inhibiting NLRP3 ATPase activity leading to the suppression of NLRP3 inflammasome activation, as previously observed in the in vitro assays.

The manuscript has some editing problems that could be easy corrected applying the mdpi styles in the whole the document”.

We have checked the manuscript to correct the editing problems.

In the attachement file, point by point response to all reviewers is included.

Reviewer 3 Report

The manuscript “Dehydrohispanolone-derived diterpene as a promising NLRP3 inhibitor agent. Bioevaluation and Molecular Docking” fits the journal’s scope. The authors present the results on dehydroisohispanolone modulation of NLRP3 inflammasome, using J774A.1 murine macrophage cells. The design of the research is clear and the experiments are well justified. The lack of a positive control (such as CY-09 or MCC950), should be compensate with by adding a proper justification.

The methods are described in sufficient detail and results are clearly presented. The discussion section regarding the docking studies should be expanded.

The conclusions are sustained by the authors’ findings. 

The separation method from Section Preparation of deshydroisohispanolone (DIH) (Supplementary material) should be presented in more detail.

Author Response

The lack of a positive control (such as CY-09 or MCC950), should be compensate with by adding a proper justification.

According to the reviewer´s suggestion, we have included additional data related to the use of MCC950 as NLRP3 inflammasome specific inhibitor.

The involvement of NLRP3 in IL-1β activation in J774A.1 murine macrophage cells has been previously reported in Gonzalez-Cofrade L et al. J. Nat. Prod. 2020. In this paper, we showed that MCC950 treatment of J774A.1 murine macrophage cells exposed to LPS and nigericin, inhibited  IL-1β activation, corroborating the role of NLRP3.

In the present study, MCC950 has also been used in parallel with the diterpene DIH as a positive control for NLRP3 inflammasome inhibition. Experiments have been performed in LPS-primed J774A.1 macrophages and BMDM following by stimulation with nigericin in presence of DIH or MCC950. IL-1β and LDH release as well as caspase-1 activity have been measured as described in materials and methods. Results have been included as a new supplementary figure S3 Moreover, we have modified the manuscript according to these new results.

The discussion section regarding the docking studies should be expanded. The separation method from Section Preparation of deshydroisohispanolone (DIH) (Supplementary material) should be presented in more detail.

Additionally, we have expanded the discussion section regarding the docking studies. As another reviewer suggested, additional covalent dockings have been carried out, discussion about the obtained results and a comparison study with the MCC950 inhibitor is included in the revised version.  Furthermore, more details about the preparation of deshydroisohispanolone (DIH) and separation have been given in this version.

In the attachement file, point by point response to all reviewers is included.

Round 2

Reviewer 2 Report

the authors made the suggested experiment and improved their paper.